



# Daily satellite-based sunshine duration estimates over Brazil: Validation and inter-comparison

Maria Lívia Lins Mattos Gava[1], Simone Marilene Sievert da Costa Coelho[1], and Anthony Carlos Silva Porfírio[2]

National Institute for Space Research Cachoeira Paulista, SP, Brazil
Research Institute for Meteorology and Water Resources Fortaleza, CE, Brazil

**Correspondence:** Maria Lívia Lins Mattos Gava (maria.gava@inpe.br)

**Abstract.**

The broad geographical coverage and high temporal and spatial resolution of geostationary satellite data provide an excellent opportunity to collect information on variables whose spatial distribution and temporal variability are not adequately represented by the in situ networks. This study focuses on assessing the effectiveness of two geostationary satellite-based sunshine duration (SDU) datasets over Brazil, given the relevance of SDU to various fields, such as agriculture and energy sectors, to ensure reliable SDU data over the country. The analyzed datasets are the operational products provided by the Satellite Application Facility on Climate Monitoring (CMSAF), that uses data achieved with the Meteorological Satellite (Meteosat) series, and by the Satellite and Meteorological Sensors Divison of the National Institute for Space Research (DISSM/INPE), that employs Geostationary Operational Environmental Satellite (GOES) data. The analyzed period ranges from September 2013 to December 2017. The mean bias error (MBE), mean absolute error (MAE), root mean squared error (RMSE), correlation coefficient ($r$) and scatterplots between satellite products and in situ daily SDU measurements provided by the National Institute of Meteorology (INMET) were used to access the products performance. They were calculated on a monthly basis and grouped into climate regions. The statistical parameters exhibited a uniform spatial distribution, indicating homogeneity within a given region. Except for the Tropical Northeast Oriental (TNO) region, there were no significant seasonal dependencies observed. The Mean Bias Error (MBE) values for both satellite products were generally low across most regions in Brazil, mainly between 0 and 1 hour. The correlation coefficient ($r$) results indicated a strong agreement between the estimated values and the observed data, with an overall $r$ value exceeding 0.8. Nevertheless, there were notable discrepancies in specific areas. The CMSAF product showed a tendency to overestimate observations in the TNO region, with MBE consistently exceeding 1 hour for all months, while the DISSM product exhibited a negative gradient of MBE values in the west-east direction, in the northern portion of Brazil. The scatterplots for the TNO region revealed that the underestimation pattern observed in the DISSM product was influenced by the sky condition, with more accurate estimations observed under cloudy skies. Additional analysis suggested that the biases observed might be attributed to the misrepresentation of clear-sky reflectance. In the case of the CMSAF product, the overestimation tendency observed in the TNO region appeared to be a result of systematic underestimation of the Effective Cloud Albedo. The findings indicated that both satellite-based SDU products generally exhibited good agreement with the ground observations across Brazil, although their performance varied across different regions and seasons.





The analyzed operational satellite products present a reliable source of data to several applications, being an asset due to its high spatial resolution and low time latency.

## 1 Introduction

Sunshine duration (SDU) is conceptually defined as the total of hours that sunlight reaches the Earth's surface directly from the sun. With the advance of the technology and the measurement instruments, it was formally defined as the sum of the periods in which direct solar irradiance reaches or exceeds 120 $W.m^{-2}$ (WMO, 2008). Along with precipitation and surface air temperature, it is one of the most important parameters in climate monitoring (Kothe et al., 2013). In a given area, the amount of sunshine received is the major factor determining the local climate (Bertrand et al., 2013).

The SDU importance has been known for a long time and its first measurements dates back to the 19th century. In fact, there are time series as long as 100 years of SDU measurements accumulated at networks all over the world. SDU data is relevant for a number applications, such as yield planning in agriculture (Rao et al., 1998; Huang et al., 2012; Wang et al., 2015), analysis of the thermal loads and sunshine duration on buildings (Shao, 1990), input parameter in soil water balance models (Warnant et al., 1994), and even research on human's health (McGrath et al., 2002; Nastos and Matzarakis, 2006; Keller et al., 2019).

Akinoglu (2008) stated that SDU data is the best long term, trustworthy, and readily available measurement to estimate the surface solar irradiation, due to the linear relationship between these variables, described by Angstrom (1924).

Based on that, it is clear the necessity of SDU records. However, there is a relatively small number of stations that measure it. Overall, networks of SDU are sparse and insufficient to cover large areas (Kandirmaz and Kaba, 2014). In Brazil, the National Institute of Meteorology (INMET) currently operates a network with 245 stations, besides that, it provides SDU time series

from other 85 sites, that are no longer operational. The effort to maintain this network is essential to provide reliable SDU records for climate studies. Nonetheless, comparatively to the temperature or precipitation networks with thousand stations, the SDU current network seems inadequate to cover the Brazilian large territory.

Along with that, once that meteorological station measurements are point-based observations (Zhu et al., 2020), SDU in the vicinity has to be obtained through interpolation techiniques. Therefore the accuracy of the method strongly relies primarily

upon the number and spatial distribution of meteorological stations. Generally, the station's distribution is heterogeneous, with most of them near cities, and for several reasons, extensive areas remain without records. For instance, it can be observed in Brazil, where some regions as the Northern have very few stations, while the South and Southeast regions present a denser network. Consequently, the resulting interpolated field is usually poor for representing the temporal and spatial SDU variability characteristics (Wu et al., 2016).

Geostationary satellites perform measurements with high spatial and temporal resolution and cover large areas, so their data can be used as an alternative to estimate SDU. Literature on the subject provides some proposed methods to accomplishs





it. Given the fact that clouds are the primary responsible for SDU changes, several methods to derive SDU rely on it. Different techniques have been considered, such as SDU estimate based on cloud cover index (Kandirmaz, 2006; Ceballos and Rodrigues, 2008; Shamim et al., 2012) and from satellite-derived cloud-type products (Good, 2010; Wu et al., 2016; Zhu et al.,

2020). Another approach is given by Kothe et al. (2017), in which SDU is calculated based on direct normal radiation threshold, using data from the Meteorological Satellite (Meteosat) series operated by European Organization for the Exploitation of Meteorological Satellites (EUMETSAT). Currently, this approach has been considered one of the most advanced remote sensing tool to estimate the SDU, being operational at the Satellite Application Facility on Climate Monitoring (CMSAF).

In the South America, Ceballos and Rodrigues (2008) proposed a SDU estimation method based on Geostationary Opera-
tional Environmental Satellite (GOES) data. It is operational at the Satellite and Meteorological Sensors Divison in the National Institute for Space Research (DISSM/INPE). This method (hereafter DISSM method) was validated by the authors using in situ measurements from São Paulo and Fortaleza cities, and their results indicated a good agreement between the estimates and the observed data. Thereafter, Porfirio (2012) extended the validation of INPE method for the Northeast Brazil, using records from 53 stations for 2008, showing a good performance of the method. Notwithstanding, the climatic characteristics of that
region are not representative of the whole country.

In addition, the satellite-derived SDU data set from CMSAF (hereafter CMSAF method) also provides estimates over Brazilian territory. However, the accuracy of the product was only evaluated for the monthly sums, and regarding few stations Kothe et al. (2017). Therefore, it is still needed a deep validation of daily SDU estimates based on satellite data over Brazil.

In this study we aimed to evaluate two operational SDU estimates products' performance over Brazil and inter-compare their
results, the following scientific questions were addressed:

1. Are those products a good fit to the in situ measurements?

2. There are regions where one of the products performs better than the other?

3. There are seasonal variations in its performance?

4. Deficiencies in the retrieval can be traced to a source?

The manuscript is structured as follows: Section 2 gives a description of the operational algorithms evaluated, i.e. the DISSM and CMSAF methods. Section 3 describes the ground measurements and statistical metrics used for the validation. Section 4 and 5 presents the obtained results and conclusions, respectively.

## 2 Satellite-based methods

### 2.1 DISSM

The DISSM has the mission to produce high quality satellite products and thus to offer relevant information for different Brazilian sectors. The DISSM was instituted in 2020, and incorporated the former Satellite Division and Environmental Systems (DSA), that operated since 1986. The division established itself as a reference in collecting and processing satellite data





and product generation for South America (Costa et al., 2018). Among their products are sea surface temperature, severe weather monitoring, precipitation estimation, sunshine duration and several others (see http://satelite.cptec.inpe.br/ and Costa et al. (2018).

To estimate SDU, visible imagery acquired with GOES processed by the DISSM are used. From time to time the GOES platforms are replaced, during the analysed period in this study (2013-2017), GOES-13 was operational and carried the IMAGER sensor on board. The IMAGER visible channel is centered at 0,65 $\mu$m, with a bandwidth of 0,2 $\mu$m.

The sensor measures the spectral radiance $L_\lambda$ ($W.m^{-2}.sr^{-1}.\mu m^{-1}$), that represents the mean value at the pixel area. The

95 spectral irradiance at the top of atmosphere is $E_o = \mu S_\lambda$, where $S_\lambda$ is the solar irradiance at normal incidence in this same spectral interval, and $\mu = cos(SZA)$ is the cosine of the solar zenith angle. Assuming that the reflected radiance is isotropic, the emergent spectral irradiance at the top of atmosphere is $E \uparrow = \pi L_\lambda$ and the reflectance is $R = \frac{E\uparrow}{E_o}$. Operationally, from the satellite visible imagery the reflectance factor (F) and the planetary reflectance (R) are defined as showed in Eq. (1) (Ceballos and Rodrigues, 2008):

$$F = \pi \frac{L_\lambda}{S_\lambda}; R = f \frac{F}{\mu} \qquad (1)$$

where the factor $f$ is a function correcting the effects of anisotropic reflection (Lubin and Weber, 1995), for the following purposes, $f$ is considered 1 (Ceballos et al., 2004). The $R$ is provided by DISSM as a by-product of the operational processing of the GL1.2 shortwave radiation model (Ceballos et al., 2004).

It is usual to regard the reflectance as a mean value between the cloud reflectance ($R_{max}$) and the clear sky reflectance

($R_{min}$) weighted by the fraction of the pixel covered by clouds (C) as showed in Eq. (2) (Ceballos et al., 2004).

$$R = C.R_{max} + (1 - C).R_{min} \qquad (2)$$

which leads to an estimate of cloudiness (C) as:

$$C = \frac{R - R_{min}}{R_{max} - R_{min}} \qquad (3)$$

Ceballos et al. (2004) defined the value of $R_{max}$ as 0.465, which corresponds to the transition between a cumuliform and a

110 stratiform cloud field and the $R_{min}$ as 0.09, a reasonable value for continental surface. In the case of $R < R_{min}$, C is set as 0, and if $R > R_{max}$, C=1. In case of $R = 0$ or marked as "invalid" (i.e. $R = -99$), $C$ is also tagged as invalid.

Next, assuming that the average cloud cover assessed by C is also representative of the relative time of cloud passage over a site inside the pixel (Porfirio and Ceballos, 2017), 1-C corresponds to the relative time of clear sky. The daily SDU is achieved through the Eq. (4), that is similar to the integration via trapezoidal rule (which consists of a numerical method to approximate

the integral value):

$$SDU = (1 - C_1) + \frac{\Delta t}{2}[(1 - C_1) + 2(1 - C_2) + 2(1 - C_3) + 2(1 - C_{k-1}) + (1 - C_k)] + (1 - C_k) \qquad (4)$$



where $C$ is the cloudiness parameter (described in Equation 3), $C_1$ corresponds to the first "valid" observation for the pixel, the subscript index corresponds to the number of the image within a day, $k$ is the last valid image of the day, and $\Delta t$ is the time interval between two consecutive images (for the period analyzed, usually 30 minutes).

On average, for a given pixel, 30 images are available for the daily SDU estimate. However, this value can be smaller and the interval between two consecutive images can be larger than 30 minutes. The daily SDU for a given day is considered invalid, therefore, discarded, if there is a interval greater than three hours: i) between the first image of the day and the sunrise; ii) between consecutive images; and iii) the last image of the day and the sunset and if less then 5 images were available for the estimation.

The spatial resolution of the DISSM's SDU dataset is 0.04 degrees on a regular latitude-longitude grid of 1800x1800 pixels within latitudes 50° S to 21.96° N and longitudes 100° W to 28.04° W, and cover the time period from February 2007 to near real time.

## 2.2 CMSAF

The EUMETSAT's CMSAF was established to contribute to the operational monitoring of the climate and the detection of
130 global climatic changes. With this aim CMSAF's products follow the highest standards and guidelines as lined out by Global Climate Observing System (GCOS) for the satellite data processing.

The SDU is one of the several products of CMSAF based on Surface Radiation Data Set - Heliosat (SARAH) - Edition 2.1 (Pfeifroth et al., 2019). The data record covers the time period from 1983 to near real time with a spatial resolution of $0.05° \times 0.05°$. In order to derive the SARAH-2 surface parameters, the Heliosat algorithm is used (Hammer et al., 2003). It
provides a continuous dataset of Effective Cloud Albedo and minimizes the impacts of satellite changes and artificial trends due to degradation of satellite instruments through an integrated self-calibration parameter Mueller et al. (2011).

At first, the Effective Cloud Albedo is retrieved by the normalized relation between all sky and clear sky reflection in the visible channel of the Meteosat instruments. This parameter is used to derive cloud index, a measure of the impact of the clouds on the clear sky irradiance. The SPECMAGIC model is used to estimate clear sky irradiance, and then from the combination
of cloud index and clear sky irradiance, Surface Incoming Shortwave radiation (SIS) is achieved. Thereafter using the diffuse radiation model of Skartveit et al. (1998) and the cloud index, Surface Incoming Direct radiation (SID) is calculated. By normalizing it with the cosine of the solar zenith angle, the DNI is obtained. The SID and DNI are the basis for the SDU estimates (Kothe et al., 2017).

Daily SDU is calculated as the ratio of slots exceeding the DNI threshold, considered as sunny slots, to all slots during
daylight (Eq. (5)).

$$SDU = daylength \times \frac{\sum_{i=1}^{daylightslots} W_i}{daylightslots} \qquad (5)$$





The day length is calculated depending on the date, longitude and latitude and is restricted by a threshold of the solar elevation angle of 2.5° (Kothe et al., 2013). $W_i$ is a weight that varies between 0 and 1, and indicates the influence of a single slot depending on the number of surrounding cloudy and sunny grid points (Kothe et al., 2017).

A grid point at a time slot $i$ is accounted as sunny if DNI is 120 $W.m^{-2}$ or larger (Equation 6). Since SARAH-2 provides instantaneous DNI data every 30 minutes, without weighting, one sunny slot would correspond to a 30 min time window. This is not the case unless it is a bright weather situation. If there are clouds in the vicinity of a grid point, probably not the whole 30 minutes are sunny. The opposite case is also valid. To account this fact, the information of the 24 surrounding grid points (Figure 1) and two successive time steps are used (Kothe et al., 2017).

$$SIn_i = \begin{cases} 1 \; if \; DNI(x,y) \geq 120 W.m^{-2} \\ 0 \; if \; DNI(x,y) < 120 W.m^{-2} \end{cases} \tag{6}$$

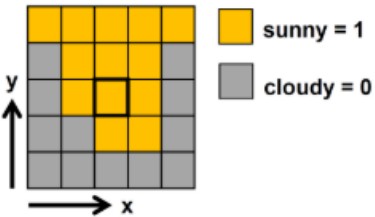

Source: Kothe et al. (2017).

**Figure 1.** Demonstration for accounting for surrounding grid points. The target grid point is marked in the center.

For each grid point, the number of sunny slots in the 24 pixels in the vicinity plus the center cell grid of interest is summed up (Equation 7).

$$\#SIn_i(x,y) = \sum_{m=y-2}^{m=y+2} \sum_{n=x-2}^{n=x+2} SIn_i(m,n) \tag{7}$$

First, for each daytime slot $i$ this is done. Then to also incorporate the temporal shift of clouds, the number of each time step

is combined with the number of the previous time step for each pixel (Equation 8).

$$N_1 = \#SIn_1 \times 0.04$$
$$N_i = (\#SIn_i + \#SIn_{i-1}) \times 0.02 \tag{8}$$

The factor 0.04 is used for the first time slot of the day, for $i > 1$, 0.02 is used, thus if all 25 grid points are sunny the resulting number N is 1, and 0 in the case that no grid point is sunny.



Thereafter the impact of sunny and cloudy grid points on the temporal length of one time slot is estimated. The fraction of
time, which slot $i$ accounts to the daily SDU is achieved by Equation 9.

$$W_i = \begin{cases} max(N_i, C_1) \; if \; DNI(C_{gp}) \geq 120 \, W.m^{-2} \\ N_i \cdot C_2 \; if \; DNI(C_{gp}) < 120 \, W.m^{-2} \end{cases} \tag{9}$$

If the DNI in the center grid point is equal or greater than $120 W.m^{-2}$, the grid point is taken as sunny, and the weight ($W_i$) is
defined by the maximum value between $N_i$ and $C_1$. Otherwise, $W_i$ is the product of $N_i$ and $C_2$. These constants were derived
empirically through sensitivity tests, by minimizing the bias compared to reference station data in Germany. They are set as
$C_1 = 0.4$, that indicates the minimum fraction that a sunny slot can contribute, and $C_2 = 0.05$, the weight for the contribution
of a non-sunny slot (Kothe et al., 2017).

The daily SDU in hours is then derived by Eq. (5).

## 3    Ground data and evaluation methods

To evaluate the satellite products for the period from September 2013 to December 2017 data from INMET's network were
used as ground "truth". This network provides daily measurements obtained with Campbell-Stokes recorders. The stations
were selected based on data availability: only stations with less than 20% of missing observations over the study period were
retained. A total of 193 stations were chosen. Quality checks were applied in order to exclude spurious data, e.g. measurements
higher than its physical limits and false zeros (Gava, 2021).

Due to the considerable extension of the Brazilian territory, as well as the great variety of biomes and climates, the stations
were grouped by climate zones, as suggested in Raichijk (2012). This classification was developed by the Brazilian Institute
of Geography and Statistics (IBGE) and takes into account the average air temperature and precipitation regimes (highly
anticorrelated to sunshine duration). The regions are illustrated in Figure 2.

Figure 3 presents the boxplot of SDU ground measurements for the analysed period grouped by the climate zones. The main
characteristics of the regions are described below, indicating inside parentheses the number of stations included in each one.

- Equatorial region (EQ, 27(*)): has Af/Am climate according to the Köppen-Geiger classification, it presents average
annual temperatures between 24 and 27°C, and average annual precipitation of 2300mm. Presents a short dry season in
winter, lasting under 3 months (Raichijk, 2012). The average SDU values ranges from 4.5 h in the wet season to 8.5 h in
the dry period, which shows smaller variability.

    (*) For the CMSAF evaluation, 22 stations were used, since 5 of the listed stations are out of the METEOSAT disk.

- Tropical Equatorial region (TE, 43): with Aw/BSh climate according to the Köppen-Geiger classification. Hot, semi-arid
with a prolonged dry season (over 8 months). This region comprises the northeast Brazilian Sertão, going south until
approximately 10°S (Raichijk, 2012). Presents high SDU values, on average, over the whole year. With highest values
in the winter, along with the smallest variability, indicating predominance of clear sky days.



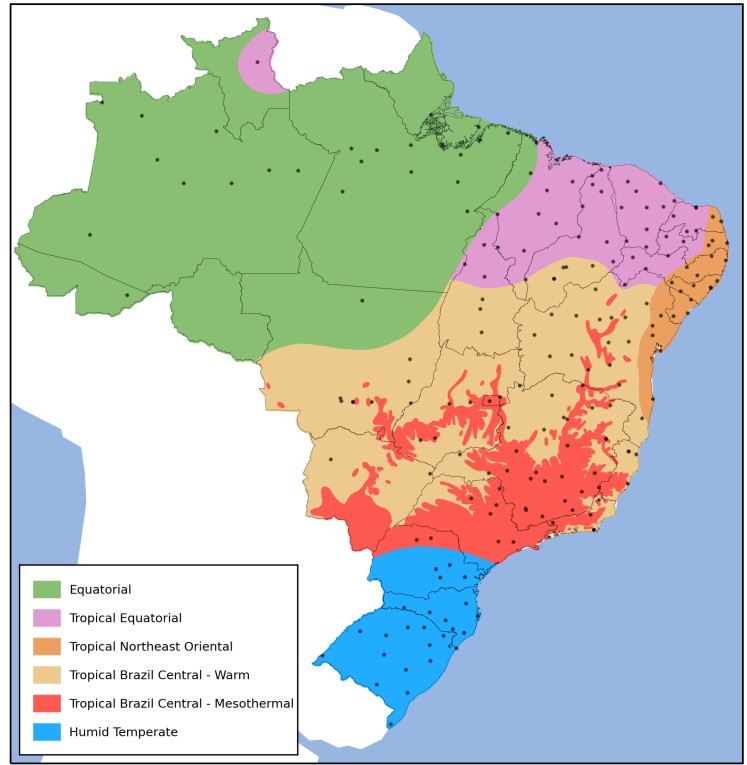

**Figure 2.** Spatial distribution of INMET's stations used in this study.

– Tropical Northeast Oriental region (TNO, 21): Mainly Aw/BSh climate according to the Köppen-Geiger classification, with average annual temperatures ranging from 24 to 26°C. The mean SDU values are high for most of the year. Smaller SDU values are found in late autumn-winter, due to the precipitation regime of this location that exhibits the maximum precipitation rates during this period (Palharini and Vila, 2017), when it also displays the highest variability along the year.

– Tropical Central Brazil region: This main region was sub-divided in a) Mesothermal/Subwarm (hereinafter, Mesothermal) and b) Warm.

– Mesothermal (TCB-M, 29): Classified as Cw climate, conform to the Köppen-Geiger classification (Raichijk, 2012). It includes part of Brazil's Southeast and north of Paraná. This region exhibits average annual temperatures between 10 and 18°C, with dry winter. It presents great SDU variability through most of the year, with the highest mean SDU values in winter.



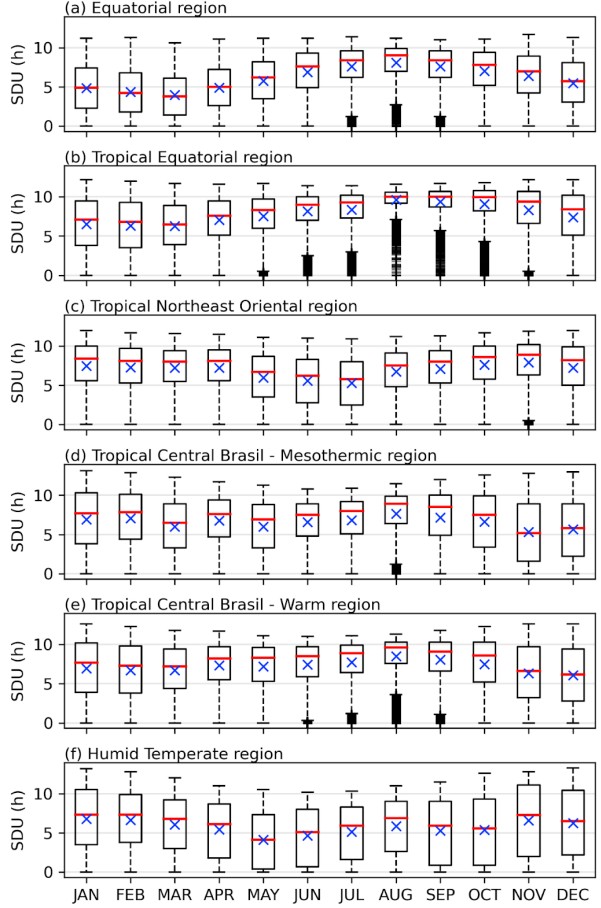

**Figure 3.** Boxplot of SDU ground measurements. The "whiskers" (lines extending parallel from the boxes) indicate variability outside the first and third quartiles. Outliers are plotted as individual crosses. The red line indicates the median and the blue 'x' the mean.

- – Warm (TCB-W, 49): Aw climate according to the Köppen-Geiger classification. This region comprises the Brazilian Central Plain. Semi-humid, marked by rainy summer and dry winter (Cavalcanti, 2009). This feature is noticeable in Figure 3. During summer, the mean SDU is smaller, with higher variability.

- – Humid Temperate region (HU, 25): It is defined as Cfa by the Köppen-Geiger classification. This region presents mild temperatures between 10 and 15°C, with a well distributed precipitation regime (without a dry season) (Raichijk, 2012). Which is perceptible in Figure 3. There is no clear seasonality in mean SDU values, moreover there is high variability in the entire year.

In order to compare the satellite-based gridded SDU estimates with the in situ records, the satellite data was extracted at the station sites, by selecting the satellite pixel in which the station is located. Therefore, for each month, the mean bias error



(MBE), mean absolute error (MAE), root mean squared error (RMSE) and the correlation coefficient (r) of the daily SDU were
calculated for every station, and then, grouped into the regions.

The definitions of the statistical measures are presented below Wilks (2011):

$$MBE = \frac{1}{n}\sum_{i=1}^{k}(Z_i - O_i)$$

$$MAE = \frac{1}{n}\sum_{i=1}^{k}|Z_i - O_i|$$

$$RMSE = \sqrt{\frac{1}{n}\sum_{i=1}^{k}(Z_i - O_i)^2}$$

$$r = \frac{\sum_{i=1}^{k}(Z_i - \bar{Z})(O_i - \bar{O})}{\sqrt{\sum_{i=1}^{k}(Z_i - \bar{Z})^2}\sqrt{\sum_{i=1}^{k}(O_i - \bar{O})^2}}$$

Thereby, the variable $Z$ describes the dataset to be validated (e.g. DISSM SDU) and $O$ denotes the reference dataset (i.e. in
situ measurements). The individual time step is marked with $i$ and $k$ is the total number of time steps.

## 4   Results and Discussion

Figure 4 displays the spatial distribution of the monthly MBE of daily SDU. It is possible to observe that the behavior of the
models is homogeneous within regions. Small bias values are noticeable for the majority of the Brazilian territory for both SDU
products, overall, ranging from -1 to 1 h. The exception is the TNO region for the CMSAF product and most of the northern
and northeast regions for the DISSM product.

On the Northeastern Brazilian coastline, the CMSAF method presents a significant tendency to overestimate the SDU values,
exhibiting high positive MBE values, reaching 4 h in some stations.

The DISSM method presents high MBE values for the EQ, TE, and TNO regions. For the first, an overestimation tendency
is observed, while the others show a negative bias. The MBE results on the TE region do not present the same homogeneity of
the other zones. The eastern portion of the region shows negative values, while the states of Maranhão and Piaui do not indicate
biases (values close to zero).

The MBE results over the Northeastern Brazilian littoral of both products presented a marked seasonality, with smaller
values in the winter. On the DISSM method, this behavior extends to the inland of the Northeast. This characteristic is evident
in Figure 5a. It can be seen that, from April to August, the DISSM's MBE approaches zero while the CMSAF's decreases
(from 1.7 h in December to 1 h). In this period, all the summary statistics present the best results: smaller MAE and RMSE
(overall, under 1.5 h and 2.0 h, respectively), and higher correlation coefficients ($r > 0.65$).



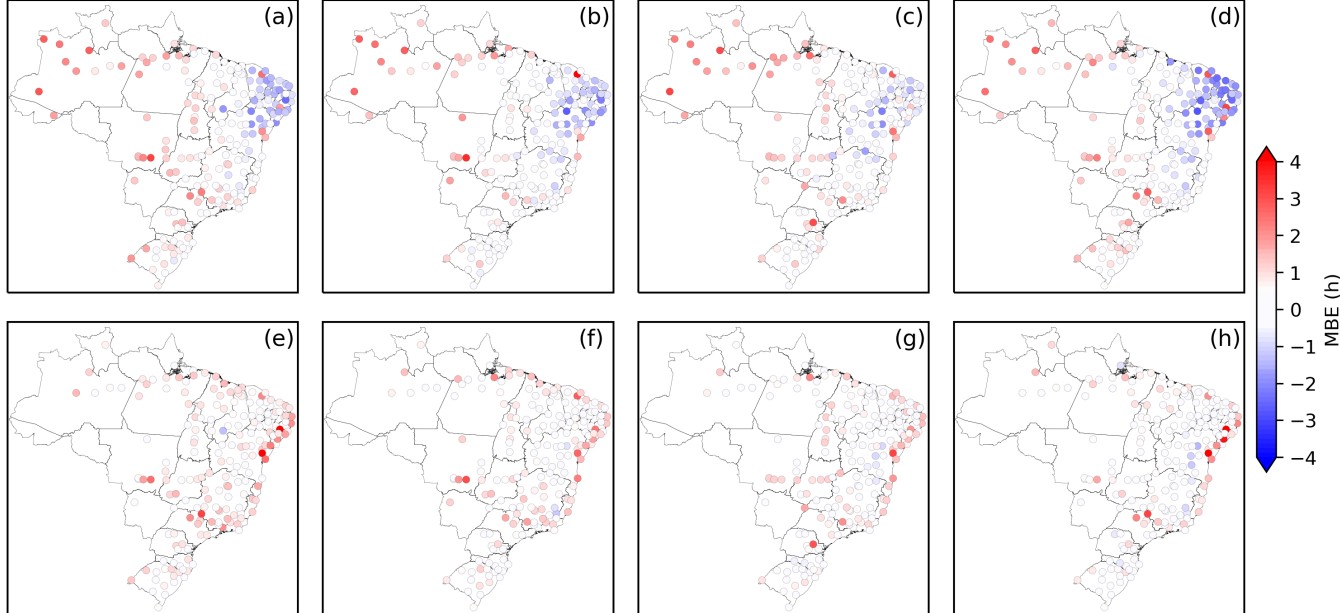

**Figure 4.** Spatial distribution of monthly MBE (h) between daily satellite-based SDU and in situ data for the period of 2013-2017. DISSM's (CMSAF's) MBE results are shown for Jan, Apr, Jul and Oct in panels a, b, c and d (e, f, g and h), respectively. Shades of red correspond to overestimation, while shades of blue correspond to underestimation.





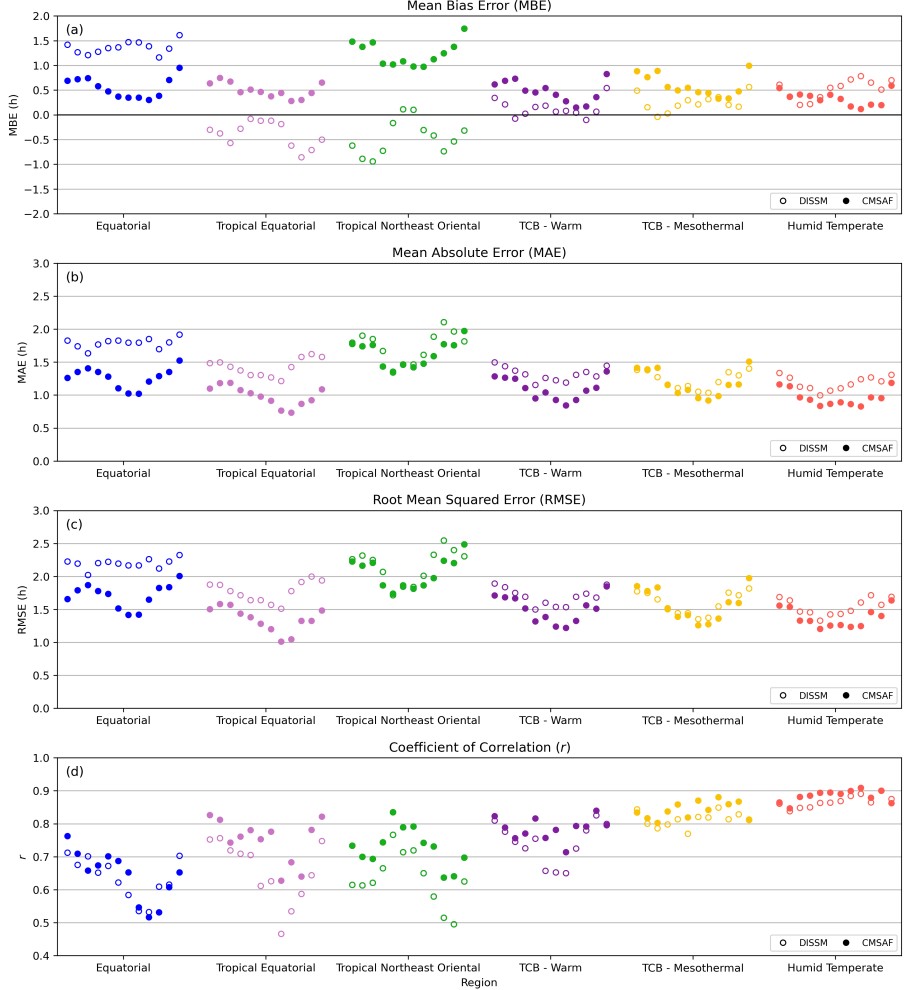

**Figure 5.** Summary statistics of the comparison between satellite-based SDU and ground data, on a monthly basis, over the Brazilian regions: (a) MBE, (b) MAE, (c) RMSE, and (d) r. The regions are plotted in different colors: EQ is in blue, TE is in lilac, TNO is in green, TCB-W is in purple, TCB-M is in yellow and HT is in red. For every region the point's x-coordinates refer to the month of the year from January through December. Open circles (filled dots) represent DISSM's (CMSAF's) results.

On the other regions, the CMSAF's MBE lies close to 0.5 h, with some variation depending on the month. The DISSM method shows small biases for the southern regions, on average, smaller than 0.5 h. On the EQ region, as previously mentioned, high positive biases are found: with its highest (smallest) value in December (October), 1.61 h (1.16 h).

Figure 5b,c,d presents the MAE, RMSE, and $r$, respectively. Those are measurements of accuracy, and together they can be used to acquire information regarding the of data spread. Considering the MAE and RMSE, it can be noticed that the CMSAF product presents mostly smaller values than the DISSM method. Over all regions, the RMSE values are higher that the MAE, indicating the presence of outliers and some dispersion of points.




On the TE and the southern regions (TCB-W/M and HT), both products exhibit the smallest errors: the MAE (RMSE) lies, typically, under 1.5 h (2 h). These areas also present high values of $r$ (on average, $r > 0.7$), indicating a good agreement between the products and the in situ data, with exception of the TE region, which presents great variability in the $r$ results, mostly for the DISSM product.

On the EQ region, although the CMSAF's MAE and RMSE results imply smaller errors compared to the the DISSM data, the $r$ results indicate the same degree of spread, varying between 0.5 and 0.8.

Over the TNO region, the products present high values of MAE and RMSE, with the above mentioned seasonality (smaller values during winter). Along with highly variable values of $r$, varying from 0.5 in November to 0.77 in May for the DISSM product, and between 0.64 in October/November and 0.83 in May for the CMSAF method. To assess the main characteristics of the products for this region, the satellite-based data was plotted against ground measurements, as well as the difference between product and observations against observations. Those scatterplots are displayed in Figure 6 and 7, for the CMSAF and DISSM products, respectively. The data is grouped by frequency of occurrence.

There is a common pattern that can be observed in both figures: an overestimation of SDU values when the ground measurements indicate zero brightness hours. This discrepancy arises from the presence of false zeros in the SDU reference time series. On days where no data is available, SDU is assigned a value of zero, resulting in these false zeros. The substantial overestimation observed highlights that the steps undertaken in the quality control process were insufficient to eliminate all erroneous data from the analysis. However, it is important to note that for non-zero values of SDU, the recorded measurements should still fall within the instrument uncertainty range of 0.1 h (Stanhill, 2003).

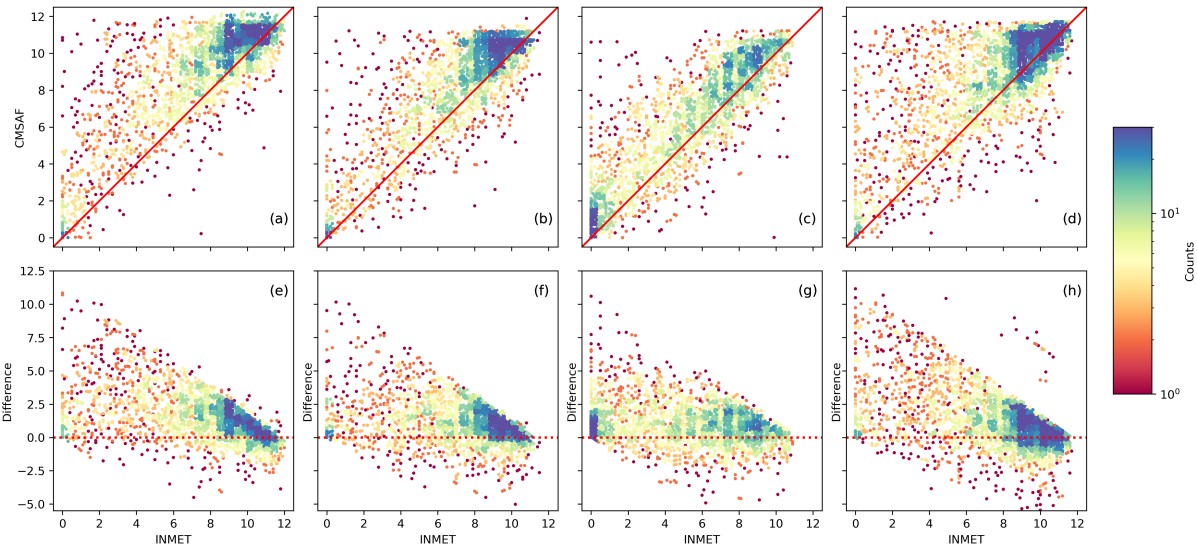

**Figure 6. Upper**: Scatter plot of the CMSAF product vs. SDU in situ data for the TNO region. **Lower**: Scatter plot of the difference between CMSAF product and SDU in situ data vs. SDU in situ data. Plots are grouped by frequency of occurrence. Boxes from left to right presents the data for January, April , July and October.





265 In Figure 6, the tendency to overestimation of the CMSAF product is evident in all analyzed months, with great dispersion above the 1:1 line. The scatterplots of the difference against observations show that under all sky conditions the product tends to present higher values than the observations. Kothe et al. (2017), when evaluating the CMSAF product found a similar behavior on the Canary Islands (Northwestern Africa), in the daily evaluation (MBE values close to 2h) and on the West coast of Africa, in the monthly sums analysis (bias values up to 50h were found). The authors attributed those uncertainties to two

270 causes: the frequent low cloud fields, predominant cloud types in these regions that cause a systematically underestimation of the Effective Cloud Albedo by the Heliosat algorithm because of the self-calibrating method (Hannak et al., 2017), leading to an overestimation of SDU; and to the constants used during the SDU estimates, which were derived empirically using data from Germany to take into account the contribution of different sky conditions to the SDU. Consequently, the presence of low warm clouds, that are most frequent over the ocean and subtropical subsidence regions (Huang et al., 2015) may not be well

275 represented by these parameters. This might be the case of the Northeast Brazil, since low clouds with relatively warm tops are the prevailing cloud type due to subsidence in the area associated with the Walker cell (Machado et al., 2014).

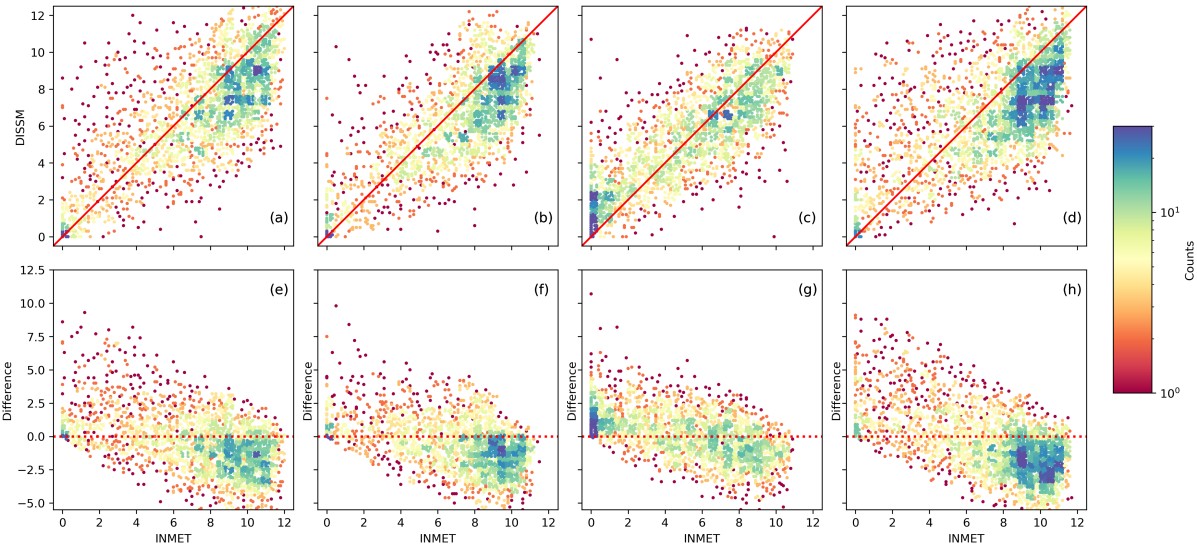

**Figure 7.** Same as Figure 6, but for the DISSM product.

 In the DISSM scatterplots' results, the aforementioned underestimation tendency is clearly seen, mostly under clear sky conditions (SDU > 8h), and is less pronounced in July. In this period, the data tend to align along the 1:1 line, which is partly due to the fact that it coincides with the rainy season in the area and ,therefore, days with less sunshine hours are more common.

280 Notwithstanding, in this particular month, for high values of SDU, the differences tend to be smaller when compared to similar values on the other months. It can be observed in Figure 7(g) that the higher frequency of points is bounded close to -2.5 h, while on other months there is a great amount of observations for differences between -2.5 and -5 h.

 The better agreement between the DISSM estimates and the observations under partly covered sky, and the higher errors in clear sky cases suggests that this might be caused by a misrepresentation of the clear-sky reflectance used in the algorithm. This



characteristic was pointed out by Porfirio et al. (2020), when evaluating the global solar irradiance (a variable highly correlated to SDU (Stanhill, 2003) estimated from GL1.2 model, the authors also found an east-west gradient in the MBE values. They highlighted the importance of a proper assessment of the clear sky reflectance to the estimation of the cloud cover, which is estimated in the GL1.2 in the same fashion as SDU product.

Figure 8 presents the mean $R_{min}$ fields for the central months of each season for the period from October 2013 to October 2017. This figure was constructed to help explore the $R_{min}$ influence on DISSM SDU product. The fields were generated by taking the minimum (not null) reflectance values for each pixel considering the available images in the interval between 14 and 16 UTC, within the target month. Thereafter, they were smoothed by performing 3x3 pixel means, then, monthly averaged.

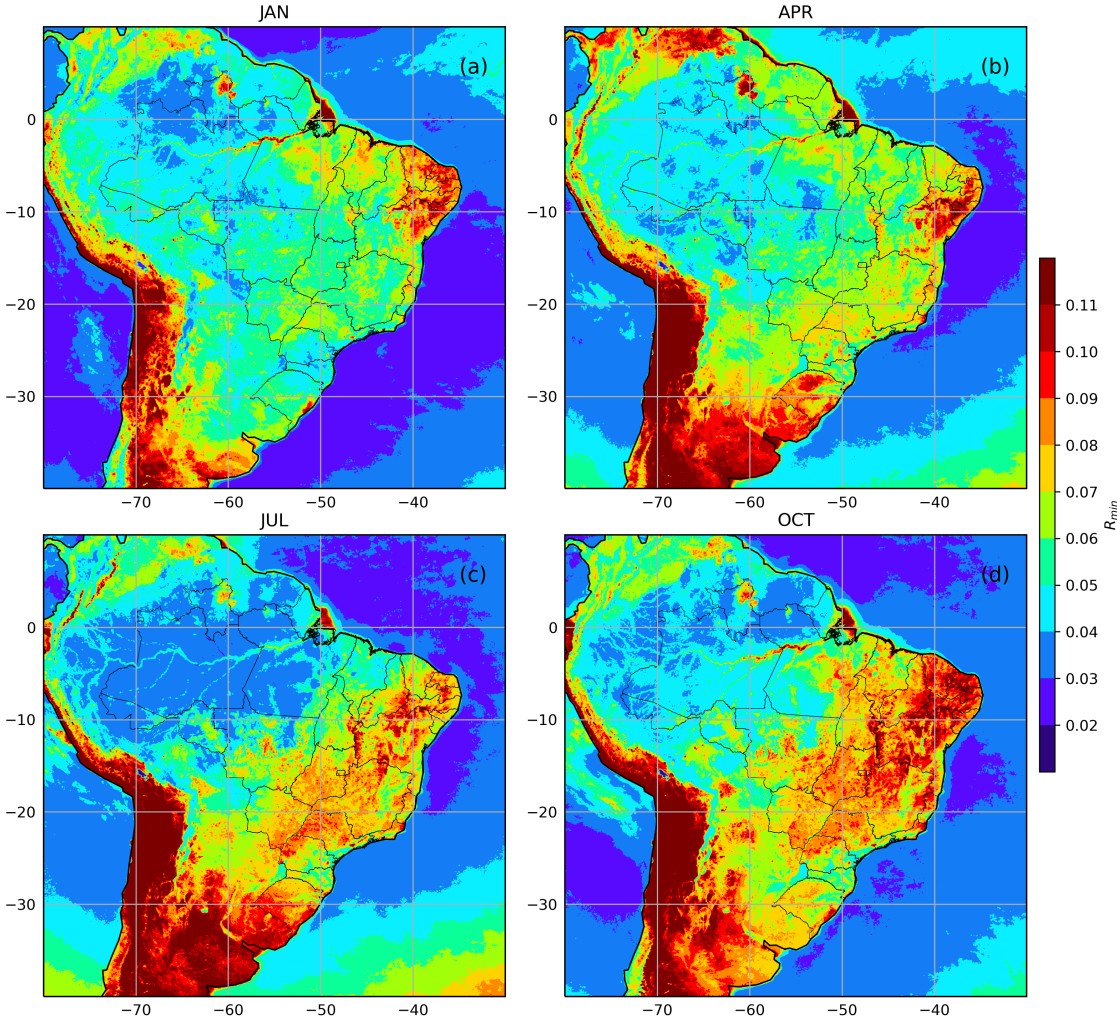

**Figure 8.** Average $R_{min}$ obtained for each analysed month in the period from Oct/2013 to Oct/2017. Blueish (Redish) colors correspond to low (high) reflectance values.



It is noticeable that there is a great variability within the country. Over Brazil, higher reflectance values are observed in Northeast Brazil and lower values in the Amazon region for all seasons and most of the regions display seasonal variations in clear sky reflectance.

Evaluating the MBE results and Figure 8, it can be seen that the $R_{min}$ has a remarkable influence on the performance of the product, particularly in the North and Northeast regions. The overestimation (underestimation) tendency found in the E (TNO) region seems to be related to clear sky reflectance values systematically lower (higher) than the $R_{min}$ used in the model (0.09). This leads to an underestimation (overestimation) of cloudiness and consequently an overestimation (underestimation) of SDU. Those results suggests that the use of fields of $R_{min}$ with spatial-temporal variations may improve the DISSM product's results.

## 5 Conclusions

The agreement between two satellite-derived SDU products and in situ measurements was accessed. The monthly bias discloses a fine agreement between SDU retrieved and observed. Both products presents low bias to most of the country (among -1 and 1 h). Exceptions emerges for the CMSAF product on the Northeastern Brazilian coastline, reaching 4 h in some stations, and for the DISSM product on the Nothern portion of Brazil, where it presents a negative gradient of MBE values in the west-east direction. The best results obtained were for southern regions (TCB-W/M and HT), with MAE (RMSE) under 1.5 h (2 h) to all months and overall high values of $r$. To the Equatorial region, the DISSM product presents an overestimation tendency through the year, with highest (smallest) MBE value in December (October), 1.61 h (1.16 h). MAE and RMSE results of the CMSAF suggest lower errors when compared to the DISSM data, despite that the values of $r$ indicate a similar level of variability, ranging from 0.5 to 0.8. To the Northeastern Brazilian coastline, both product exhibits a seasonal variation in performance, with better fit for the winter months. Although, this feature is more evident to DISSM data. During this season, both products shows smaller MAE and RMSE (generally, under 1.5 h and 2.0 h, respectively), and higher correlation coefficients ($r > 0.65$). For other periods, MAE (RMSE) varies between 1.5 and 2 h (2 and 2.5 h) and the CMSAF data generally has a higher correlation coefficient than DISSM product. In the TNO region, the systematically underestimation of the Effective Cloud Albedo due to the low warm clouds might be the cause to the observed SDU overestimation of the CMSAF product. The scatterplot analysis indicated that the DISSM's underestimation tendency over the TNO region is more evident under clear sky condition (SDU > 8h), which suggests that this might be caused by a misrepresentation of the clear-sky reflectance used in the algorithm. Further investigation on the $R_{min}$ showed that, in fact, in this region, the usual values of $R_{min}$ are over the $R_{min}$ specified in the algorithm, leading to overestimation of the cloudiness parameter and consequently underestimation of SDU. The opposite is observed on the Equatorial region, where the $R_{min}$ is usually under the algorithm's parameter. This suggests that including the spatial-temporal variations of $R_{min}$ in the DISSM algorithm may improve its performance. All in all, the results obtained in this study suggests that both products are adequate in providing reliable SDU data to a variety of applications.



*Data availability.* CMSAF SDU data is available at https://wui.cmsaf.eu/safira/action/viewProduktDetails?eid=21845_22003&fid=28 (Pfeifroth et al., 2019). DISSM SDU data for the period used in this study and GOES reflectance data used for producing Figure 8 are available at:

https://doi.org/10.5281/zenodo.7958199 (Gava et al., 2023b) and https://doi.org/10.5281/zenodo.7963354 (Gava et al., 2023a), respectively. For the entire period covered by the DISSM SDU product, the data can be made available upon request to DISSM/INPE. The SDU ground measurements are available at https://bdmep.inmet.gov.br (INMET, 2023).

*Author contributions.* Conceptualization, M.L.L.M.G., A.C.S.P. and S.M.S.C.c.; Data acquisition and analysis, M.L.L.M.G.; Writing—Original Draft Preparation, M.L.L.M.G.; all authors contributed to the discussion, review, and editing. All authors have read and agreed to this version

of the manuscript.

*Competing interests.* The authors declare that they have no conflict of interest.

*Acknowledgements.* This study was financed in part by the Coordenação de Aperfeiçoamento de Pessoal de Nível Superior - Brasil (CAPES) - Finance Code 001. We acknowledge CMSAF and DISSM for developing and keeping the satellite-based SDU datasets, and INMET for the recognized effort to provide ground measurements.



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
