# Peer review of "Daily satellite-based sunshine duration estimates over Brazil: Validation and inter-comparison"

_EGUsphere, 2023_

## Author Comment (AC1)

**Response to Reviewer Comments for "Daily satellite-based sunshine duration estimates over Brazil: Validation and inter-comparison"**

Gava et al., 2023

Reviewer comments are written in black text below, and our responses are written in *blue* text.

Referee #2 comments:

The reviewed study assesses the overall quality of Sunshine Duration (SDU) estimates through geostationary satellite data over Brazil. This is done by comparing CMSAF's product, composed by Meteosat measurements and DISSM/INPE's product, obtained from GOES series' data, against SDU estimates from in-situ measurements from meteorological stations. The analysis is based on usual statistical metrics, guided by a climate regions separation and leveraging data enhancements by the way the results are displayed.

**General Comments:**

1. The data presented, introduces the overall behavior of SDU products on climate regions in South America. I acknowledge the contribution of this study to the progress in satellite estimates of SDU in the region. This kind of assessment is meaningful for a broad range of economic and infrastructure activities developed in the study region.
2. The authors revisited the literature properly and brought to discussion the available measurements over the study region and their respective limitations. All the results were discussed accordingly. Additionally, the long description of the regions' aspects was relevant for this discussion.
3. Their results were presented in a clear, concise, and well-structured way. Some of the figures embedded a lot of relevant information from the way the data was presented. It was a smart choice to do so. Their interpretation was coherent and supported by the data presented. Substantial conclusions were reached in accordance with their analysis.

We sincerely appreciate the reviewer's remarks and suggestions, which helped to improve the manuscript's quality. We are glad that the reviewer considered that our paper contributes to the related research field and that the way the results were presented in the figures effectively communicates pertinent information allowing reaching significant conclusions. Please find detailed responses to each of the comments down below.

**Specific Comments:**

1. Particularly in Figures 4, 6, 7 and 8, it was not completely clear to me why just the middle month of every season of the year were shown. Is there a specific reason for this choice? It might be better to clear that out in the text.

We agree with the reviewer's comment that the center month's choice for figures display needs clarification. Which is as follows:

The choice for displaying only the center month of seasons was made due to the similarity between the overall behavior of the variables under analysis within seasons. Therefore, in order to minimize repetition, we chose only to use the central month as it would be representative of the general behavior without loss of information. Besides that, a summary of MBE behavior (Figure 4) is implied in Figure 5 for completeness.

Here is an example for Figure 4, with all months:

Figure 1: Spatial distribution of monthly MBE (h) between daily CMSAF's SDU estimates and in situ data for the period of 2013-2017.

[Figure]

Figure 2: Spatial distribution of monthly MBE (h) between daily DISSM's SDU estimates and in situ data for the period of 2013-2017.

[Figure]

We appreciate that it was pointed out that this explanation was missing in the manuscript and will add it to the revised manuscript.

2. Figures 6 and 7 are very illustrative of the products' general behavior. In most of those Difference vs. Ground-truth plots there are clusters of counts close to the 0.0 line, centered around larger SDU, which means the products present better performance with longer periods of clear sky. However, on "g" plots (mostly 6g but 7g as well), the data seems more spread and less clustered. Does that suggest that the products account for cloudiness better in July for most of the regions of interest?

We highly appreciate the reviewer's comment. We provide a clearer description of results for elucidation, along with the modifications to the manuscript to account for them.

The 'clusters' that can be observed in the right upper corner of Figures 6 and 7 (a, b and d) arise mainly due to the fact that in the TNO region for the corresponding months there is a high frequency of clear sky days (high SDU values) as can be observed in Figure 3. So, most of the counts are concentrated above 8 hours for the observed data (INMET, in the x-coordinate).

For the Figures 6g and 7 g, the difference behavior is seen mostly due to the higher occurrence of intermediary SDU values in this month. This is illustrated in Figure 3c: For July, it shows that the mean SDU value is lower than for other months and presents a much higher variability, which is associated with higher frequency of cloudiness in the region. This translates into a spread of counts along the x-coordinate.

These figures were thought to highlight the over/underestimation behavior of the satellite products, with the aid of the Difference vs. Ground-truth.

For the CMSAF product, the higher count's frequency is generally above the 0 line in all analyzed months, indicating that for all sky conditions the product tends to present higher values than the observations.

For the DISSM product, the higher count's frequency widely falls below the zero line, indicating the product's tendency to underestimate observation values. The exception is July (Figure 7 (g)), where the counts tend to group along to the zero line for low and intermediary values of SDU (SDU < 8 h). Although it still presents a tendency to underestimate the SDU for clear sky conditions, the errors seem to be bounded close to -2.5 h. These results are further discussed with the aid of Figure 8.

Manuscript changes:

We add the following in Line 264 (p. 14):

Figures 6 and 7 (a, b and d) present a high count's frequency in the right upper corner mainly due to the fact that in the TNO region for the corresponding months there is a high frequency of clear sky days as can be observed in Figure 3. So, most of the counts are concentrated above 8 hours for the observed data.

For Figures 6 and 7 (g), a different behavior is seen, mostly due to the higher occurrence of intermediary SDU values in this month. This is illustrated in Figure 3 (c): For July, it shows that the mean SDU value is lower than that of other months and exhibits significantly higher variability. This higher variability is associated with a greater frequency of cloudiness in the region due to the rainy season. This translates into a spread of counts along the x-coordinate.

We modified Line 265 (p. 14):

From:

In Figure 6, the tendency to overestimation of the CMSAF product is evident in all analyzed months, with great dispersion above the 1:1 line. The scatterplots of the difference against observations show that under all sky conditions the product tends to present higher values than the observations.

To:

In Figure 6, it is evident that CMSAF product tends to overestimate the SDU for all analyzed months, under all sky conditions. This is particularly clear from the scatterplots of the difference against observations, as the majority of counts tend to fall above the 0.0 line.

We modified Line 277 (p. 14):

From:

In the DISSM scatterplots' results, the aforementioned underestimation tendency is clearly seen, mostly under clear sky conditions (SDU > 8h), and is less pronounced in July. In this period, the data tend to align along the 1:1 line, which is partly due to the fact that it coincides with the rainy season in the area and ,therefore, days with less sunshine hours are more common. Notwithstanding, in this particular month, for high values of SDU, the differences tend to be smaller when compared to similar values on the other months. It can be observed in Figure 7(g) that the higher frequency of points is bounded close to -2.5 h, while on other months there is a great amount of observations for differences between -2.5 and -5 h.

To:

In the DISSM scatterplots' results, the aforementioned underestimation tendency is clearly seen, mostly under clear sky conditions (SDU > 8h), and is less pronounced in July. In this month, it is noticeable from Figure 7 (g), that the counts tend to group along to the zero line for low and intermediary values of SDU (SDU < 8 h). For clear sky conditions, although it still presents a tendency to underestimate the observations, the differences tend to be smaller when compared to similar values on the other months. The higher frequency of points is bounded close to -2.5 h, while in other months there is a great amount of counts for differences between -2.5 and -5 h.

**Technical Corrections:**

2: "**Are there** regions in which… ?"

The questions [2 - 4] after line 75 should be re-written for correctness. I understand that there would be no loss in meaning with those changes.

3: "**Are there** seasonal variations… ?"

4: "**Could** deficiencies in the retrieval be traced back **to their** source?"

The above modifications were made. We appreciate the reviewer for bringing it to our attention.

Line 173 (p.7) could be incorporated to the previous paragraph.

If the DNI (...) of a non-sunny slot (Kothe et al., 2017). **The daily SDU in hours is then derived by Eq. (5)**

Done.

Presents high SDU values, on average, over the whole year. With highest values in the winter, along with the smallest variability, indicating predominance of clear sky days

Line 193 (p. 7). Suggestion for re-writing:

→ On average, this region presents high SDU over the whole year, compared to the others. In the winter, the highest values are reached and the variability is low, due to the predominance of clear sky days.

We did the following modification to incorporate the reviewer suggestion:

On average, this region presents high SDU over the whole year. From June to September, the highest values are reached and the variability is low, indicating the predominance of clear sky days.

Figure 4. Spatial distribution of monthly MBE **(h)** between (...)

Figura 4's legend. Suggestion for avoiding confusion:

→ Figure 4. Spatial distribution of monthly MBE (hours) between (...)

We agree that it could be misleading to the readers as there is a panel (h) in the Figure, so we have incorporated the suggestion in the revised manuscript.

---

## Author Comment (AC2)

**Response to Reviewer Comments for "Daily satellite-based sunshine duration estimates over Brazil: Validation and inter-comparison"**

Gava et al., 2023

Reviewer comments are written in black text below, and our responses are written in *blue* text.

Referee #1 comments:

This study evaluates two satellite-based data records of sunshine duration for the region of Brazil. The first product is a Meteosat-based climate data record by CM SAF, the other product is a GOES-based product by DISSM / INPE. Both datasets are compared against in-situ measurements from ground stations all over Brazil. The analyses is done for different climatic regions and seasons.

**General comments**

- Sunshine duration is an important parameter as there are long-time series of measurements and it is easy to communicate to the general public, as this parameter is well known and easy to understand. This study helps to estimate the quality of satellite-based sunshine duration and shows that satellite-based products can be a good complement to station-based data.
- The study is very well structured and written. It gives an excellent overview about the two different retrieval techniques used to derive sunshine duration from satellite. The applied approach is clearly explained and the results are easy to understand and well presented by meaningful figures. Results and figures support the conclusions in an adequate way.

We deeply appreciate the reviewer's valuable comments and suggestions, which helped us enhance the quality of the manuscript. We are glad for the reviewer's mostly positive comments on our paper. We carefully considered each comment, and responded to each one down below.

**Specific comments**

- It might be good to include a paragraph describing some of the specific challenges, which have to be considered while comparing satellite-data to station-based data. For the

Meteosat-based product Brazil is right at the outer edge of the observed region, which comes along with larger uncertainties in the data due to different effects (parallax effect, larger footprint, different atmospheric effects, …). Thus, the observing geometry is different to a GOES-based product. How representative are station-based measurements for a grid point of 4 km by 4 km (or even larger for the CM SAF product at the edge of the Meteosat disc)?

We highly value this comment. We have included in the conclusions a sentence on the challenges of comparing satellite (area measurements) and station data (point measurements). We believe that the inclusion of this will help readers understand the underlying limitations of satellite pixel and station data comparison.

Added on Line 309:

"There are some challenges in comparing satellite-based products and station data. Satellite-derived observations are area measurements, and station records are point data; therefore, some representativeness error is expected due to the inherent differing spatial scale. Nevertheless, monthly bias discloses a fine agreement between SDU retrieved and observed. Both products present low bias for most of the country (between -1 and 1 h)."

Additionally, we acknowledge that despite the METEOSAT's observing geometry challenge over the analyzed region, CMSAF's product has fine agreement with ground data.

Added on Line 320:

"CMSAF's performance is remarkable, given that METEOSAT covers Brazil, particularly the EQ region, with a very high satellite viewing angle, which contributes to uncertainties on the observation due to different effects. In fact, the results indicate that the algorithm is robust and that its performance seems to be independent of the viewing angle."

- CM SAF released SARAH version 3.0 in May 2023. Especially with some improvements in underlying auxiliary data, such as water vapour and surface albedo, it is expected to perform better in the region of Brazil.

We are thankful to the referee for letting us know about this new version of SARAH. We have reviewed the algorithm theoretical basis document (ATBD) and the validation report of the new version of the dataset, and included some modifications to the dataset that might have positive impacts on CMSAF's SDU retrieval. Along with that, we included the shortened data acquisition time for the new GOES generation, which is expected to improve the DISSM's SDU estimates.

Added on Line 322:

Furthermore, it is important to highlight that both products are undergoing continuous development, thus improvements in their quality are expected. For instance, the new generation of GOES reduced the time gap between consecutive images from 30 minutes to 10 minutes; CMSAF has recently introduced an upgraded version of SARAH, which incorporates various enhancements in ancillary data, such as surface albedo, atmospheric aerosol, and water vapor profiles. These adjustments have the potential to decrease uncertainties in SDU measurements and further enhance the overall performance of these products.

**Technical corrections**

- line 206: 'summer' could be confusing for some readers, as we are on the southern hemisphere

Thank you for the comment. We agree that the terminology can be misleading, so we have changed "summer" to "austral summer" in line 207, as well as other similar occurrences in the text. Particularly for the sentence: "Semi-humid, marked by rainy summer and dry winter (Cavalcanti, 2009)" in line 206, we clarified the season period as: "Semi-humid, marked by rainy summer (December to February) and dry winter (June to August) (Cavalcanti, 2009)".

- Fig. 8: I learned that it should be avoided, wherever possible, to use rainbow colour scales. Maybe you consider this at least for the next time.

Thank you for taking the time to share your comment; it has enlightened us to the problem of using rainbow-like color schemes. We previously chose this particular color scale to facilitate comparison with Fig. 9 of Porfirio et al., 2020. However, we researched the issue of using rainbow color scale, and concluded that, indeed, this choice might be a challenge to people with color vision disabilities, besides it can mislead the interpretation of the data. So we have regenerated Fig. 8 with a different color scheme, batlow (Crameri, 2018), which was developed to prevent visual distortion of the data and exclusion of readers with color vision deficiencies (Crameri et al., 2020). We appreciate this comment as it helps us be more inclusive.

Crameri, F.: Scientific colour maps, Zenodo, 10, 2018.

Crameri, F., Shephard, G. E., and Heron, P. J.: The misuse of colour in science communication, Nature communications, 11, 5444, 20

Porfirio, A., Ceballos, J. C., Britto, J., and Costa, S.: Evaluation of Global Solar Irradiance Estimates from GL1. 2 Satellite-Based Model over Brazil Using an Extended Radiometric Network, Remote Sens., 12, 1331, https://doi.org/10.3390/rs12081331, 2020.

Figure 8. Average Rmin obtained for each analyzed month in the period from October 2013 to October 2017 Darker (Lighter) colors correspond to low (high) reflectance values.

[Figure]